# Latent Space Codon Optimization Maximizes Protein expression

## Abstract

Codon optimization, the process of selecting synonymous codons to enhance mRNA translation efficiency and protein expression, is crucial for therapeutic protein production and mRNA-based vaccines. However, it faces two major challenges: navigating a vast, discrete combinatorial space that precludes gradient-based methods, and relying on heuristic proxies like Codon Adaptation Index or GC content balancing, which often fail to capture true expression dynamics. To address these, we introduce the Latent-Space Codon Optimizer (LSCO), which reformulates the problem in a continuous latent space derived from a pretrained mRNA language model, enabling efficient gradient-based optimization. Next, LSCO incorporates a data-driven expression objective trained on mRNA-protein expression data, regularized by a Minimum Free Energy for structural stability, and employs constrained decoding to ensure mRNA-protein fidelity. Evaluated on two mRNA-protein expression dataset, LSCO outperforms baselines such as frequency-based methods and recent naturalness-driven learned codon optimizers in predicted expression yields, while maintaining structural stability and host-appropriate GC content. Our results underscore LSCO's potential in advancing codon optimization, delivering mRNA sequences that excel in expression while ensuring thermodynamic stability and organism-specific compatibility.

## 1 Introduction

Synonymous codons, three-nucleotide triplets that redundantly encode the same amino acid, create a vast design space for tailoring mRNA without altering the protein sequence (Brule & Grayhack, 2017). With 64 codons mapping to 20 amino acids, the number of possible synonymous mRNA sequences explodes combinatorially with protein length; for instance, the 1,273-amino-acid SARS-CoV-2 spike protein can be encoded by over $10^{632}$ distinct mRNAs (Zhang et al., 2021b). Within this immense space, codon selection profoundly influences translation efficiency and protein expression, where even minor synonymous changes can dramatically alter yields for industrial enzymes and therapeutic antibodies (Mauro & Chappell, 2014; Burgess-Brown et al., 2008). In applications ranging from monoclonal antibody production in cell lines to mRNA-based therapeutics, codon optimization serves as a critical tool for enhancing functional expression, underscoring the need for methods that can effectively navigate this design space (Paremskaia et al., 2024; Mauro, 2018).

Codon optimization faces two primary challenges. First, the synonymous sequence space is combinatorially vast and inherently discrete, rendering exhaustive enumeration infeasible and precluding efficient gradient-based optimization techniques. Second, the ideal objective—an accurate model of mRNA-to-protein expression efficiency—is inaccessible due to biological complexity and experimental costs, forcing reliance on heuristic proxies like Codon Adaptation Index (CAI) (Sharp & Li, 1987) or GC content balance Qian et al. (2012). These surrogates, while enabling tractable algorithms, often fail to capture the full nuances of expression, leading to suboptimal designs that may not reliably improve protein yield (Constant et al., 2023).

To address these challenges, we propose the Latent-Space Codon Optimizer (LSCO), a framework that reformulates codon optimization as a continuous, gradient-based problem while incorporating a data-driven expression objective. LSCO leverages the latent space of a pretrained mRNA language model to enable smooth optimization of codon embeddings, bypassing discrete search constraints. For the expression objective, we train a neural network predictor directly on mRNA-protein ex-

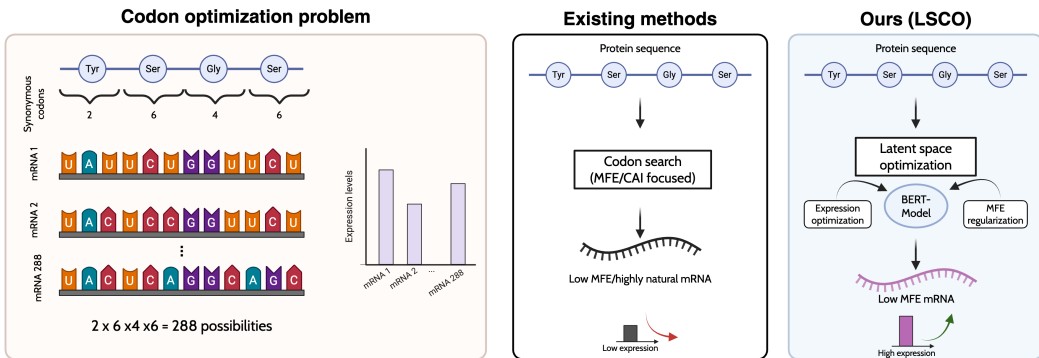

Figure 1: **Generative Codon optimization problem.** Starting from a protein sequence, we navigate the combinatorial codon design space by optimizing in a continuous latent space. LSCO integrates differentiable predictors for expression and stability, entropy regularization, and constrained decoding to produce host-compatible, high-expression mRNA sequences.

pression data to approximate mRNA-protein translation efficiency, additionally regularized by a Minimum Free Energy (MFE) to promote mRNA stability and avoid adversarial sequences. Then, the constrained decoding procedure ensures fidelity to the target protein sequence, yielding designs that balance high expression with structural integrity and host compatibility.

We evaluate our Latent-Space Codon Optimizer (LSCO) on several antibody expression datasets with experimentally measured protein expression. Across these datasets, our LSCO outperforms publicly available codon-optimization tools on expression-oriented metrics. The optimized sequences preserve amino-acid identity, exhibit predicted structural stability, and maintain host-appropriate GC content within established optimal ranges for the organism.

To sum up, we make the following contributions:

- We identify reliance on surrogate objectives and discrete search as core limitations in codon optimization and propose LSCO to jointly address them through data-driven prediction and continuous latent-space optimization.

- We detail LSCO's components, including a learned expression predictor, MFE regularization, and constrained decoding for the mRNA-protein consistency.

- On antibody expression benchmarks, LSCO achieves higher predicted yields than established methods, with stable structures and appropriate GC levels, highlighting its potential for therapeutic applications.

The remainder of this paper is organized as follows. Section 2 covers the essential related work. Section 3 provides a background on codon optimization, defining key concepts and highlighting the challenges of surrogate models and discrete search spaces. Section 4 introduces the Latent-Space Codon Optimizer (LSCO), detailing its components. Section 5 presents our experimental setup, baselines, evaluation metrics, and results on available protein expression datasets, demonstrating LSCO's performance. Finally, Section 6 discusses limitations, and outlines directions for future work.

## 2 RELATED WORK

**Codon Optimization** Codon optimization is a longstanding challenge in synthetic biology and biotechnology, aiming to enhance recombinant protein expression by adjusting gene sequences to align with host-specific translational machinery preferences. Early strategies focused heavily on maximizing the use of highly frequent codons within the host organism, leading to tools like JCat (Grote et al., 2005) and OPTIMIZER (Puigbo et al., 2007) that emphasize codon adaptation index (CAI) and codon pair bias as key metrics. However, such traditional approaches have shown

limitations, including insufficient attention to other genomic features such as mRNA stability and resulting protein expression (Zhang et al., 2021b; Demissie et al., 2025; Paremskaia et al., 2024).

Recent advances have seen the incorporation of deep learning, into the codon optimization pipeline. Models based on transformer architectures such as CodonTransformer (Fallahpour et al., 2025) use large-scale, multi-species datasets to learn complex codon usage patterns across diverse organisms. These models address organism-specific context by combining amino acid-codon pair representations and novel tokenization strategies, ensuring both species specificity and preservation of regulatory elements. Similarly, the RNop (Gong et al., 2025) framework utilizes a suite of custom loss functions targeting fidelity, tRNA availability, mRNA secondary structure, and codon adaptation within a transformer-based architecture. These advances achieve superior computational efficiency, fidelity, and improved protein expression compared to earlier methods. Comparative analyses have also highlighted the importance of integrating multiple metrics—such as CAI, GC content, mRNA folding energy, and codon pair bias—rather than relying on single-metric approaches (Zhang et al., 2021b). Multi-criteria optimization frameworks generally outperform single-objective methods in producing effective, host-specific synthetic sequences. Additionally, recent studies have leveraged recurrent neural networks and active learning to further improve translational efficiency and adaptability across expression systems (Jain et al., 2023; Li et al., 2024).

**Discrete sequence optimization** For optimizing discrete space sequences, there are two popular approaches in general: genetic algorithms and latent-space optimization. Genetic algorithms (GAs) evolve a good solution from random mutation (Deb et al., 2002) and are known being inefficient (Turner et al., 2021). Model-based genetic algorithms are used to improve the efficiency by learning a discriminative model to screen the proposed queries before labelling (Yang et al., 2019) and by learning a generative models to improve the mutation (Zhang et al., 2021a). Latent-space optimizers rely on generative models to learn a continuous representation on the latent space which is then shared with a generative decoder to reconstruct the discrete sequence and a discriminative model to predict the properties that we are interested. The output from the discriminative model will be optimized w.r.t. continuous latent representation and an optimized sequence is generated using the decoder from the optimized latent representations (Gómez-Bombarelli et al., 2018; Jin et al., 2021; Tripp et al., 2020). Latent-space Bayesian optimization can be done by replacing the discriminative property prediction model with a probabilistic one, such as Gaussian processes, which quantifies the epistemic uncertainty (Stanton et al., 2022; Gruver et al., 2023; Amin et al., 2024). This assumes an iterative process where the property prediction model will be improved by using more labeled data from future experiments; Therefore, the epistemic uncertainty is important to trade off between exploration and exploitation.

Our work differs from prior approaches by explicitly optimizing for protein expression rather than relying on proxy metrics like codon adaptation or "naturalness", employing a novel latent-space optimization framework that enables gradient-based search.

## 3 BACKGROUND ON CODON OPTIMIZATION

A protein sequence of length $L$ is denoted by $\mathbf{a} = (a_1, a_2, \ldots, a_L)$ with $a_i \in \mathcal{A}$ where $\mathcal{A}$ is the set of 20 canonical amino acids. Each amino acid $a_i$ can be encoded by a subset of synonymous codons $\mathcal{C}(a_i) \subseteq \mathcal{C}$, where $\mathcal{C}$ is the full codon alphabet consisting 64 nucleotide triplets. An mRNA sequence corresponding to $\mathbf{a}$ is thus $\mathbf{c} = (c_1, c_2, \ldots, c_L)$ with $c_i \in \mathcal{C}(a_i)$.

For algorithmic purposes, codon sequences are often represented in one-hot form $\mathbf{r} \in [0, 1]^{L \times 64}$ where each row $r_j$ is a one-hot vector indicating the codon index chosen at position $j$. We can interpret $r_j$ as a probability distribution of codons at position $j$, also permitting non-Dirac delta distributions, indicating the uncertainty of the chosen codon.

Due to the codon synonymity, protein $\mathbf{a}$ can be encoded by multiple synonymous mRNAs. The synonymous design space for given protein $\mathbf{a}$ is the combinatorial space $\mathcal{S}(\mathbf{a}) = \mathcal{C}(a_1) \times \cdots \times \mathcal{C}(a_L)$.

The codon optimization task implies parsing $\mathcal{S}(\mathbf{a})$ to find the codon sequence which maximizes expression of the protein $\mathbf{a}$, or formally:

$$\mathbf{c}^{\star} = \underset{\mathbf{c} \in \mathcal{S}(\mathbf{a})}{\arg\max}\, \Phi(\mathbf{c}) \tag{1}$$

where $E : \mathcal{S}(\mathbf{a}) \to \mathbb{R}$ denotes an mRNA-to-protein expression model that admits an mRNA sequence and evaluates its expression efficiency.

Ideally, $\Phi$ would be a perfect biophysical simulator or empirical oracle capable of predicting the exact protein yield from any given mRNA sequence, accounting for factors such as translation initiation, elongation rates, ribosomal availability, mRNA stability, and host-specific cellular dynamics (Zur & Tuller, 2016). However, such an ideal $\Phi$ is not accessible in practice due to the complexity of biological systems, incomplete understanding of underlying mechanisms, and the prohibitive cost of experimentally evaluating every possible sequence in the combinatorial design space $\mathcal{S}(\mathbf{a})$. Consequently, codon optimization algorithms rely on proxy metrics or surrogate models such as the Codon Adaptation Index (CAI), balanced GC content, mRNA stability, absence of negative motifs, and sequence naturalness Moore & Maranas (2002). These proxies enable tractable optimization algorithms, though they may not always capture the full complexity of $\Phi$. Additional challenges arise from the discrete nature of the optimization domain $\mathcal{S}(\mathbf{a})$, which precludes gradient-based optimization and necessitates sophisticated discrete search techniques Arbib et al. (2020). Finding a good proxy for mRNA-to-protein expression model $\Phi$ and searching through the combinatorially large discrete space of mRNA sequence candidates *present two major challenges of codon optimization*.

## 4 LATENT-SPACE CODON OPTIMIZATION

We propose the Latent-Space Codon Optimizer (LSCO) to simultaneously address both challenges of codon optimization. First, LSCO reformulates the discrete sequence domain into a continuous one, allowing efficient gradient-based optimization. To this end, we utilize the latent space of a pretrained mRNA language model and optimize the codon-level sequence embeddings, rather than searching the discrete mRNA space directly. At the end of optimization, we decode the optimized embeddings back to the discrete mRNA sequence. Second, instead of relying on hand-crafted heuristic surrogates such as CAI or naturalness for the expression model $\Phi$, we learn it directly from data by training a neural network to predict mRNA-to-protein expression levels, then use the predicted quantity as the optimization objective. Finally, we show that optimizing predicted expression alone can produce near-adversarial sequences with high predicted yields but poor stability or GC content. To counter this, we regularize the objective by also minimizing the Minimum Free Energy (MFE), which yields sequences that are both highly expressive and stable, with host-appropriate GC content levels. The overall LSCO framework is presented in Figure 2. With this, given the protein of interest $\mathbf{a}$, we can rewrite Equation 1 into the LSCO objective:

$$\hat{\mathbf{z}} = \underset{\mathbf{z}}{\arg\min}\, \Big\langle -\lambda_1 \Psi(\tilde{\mathbf{r}}_\tau) + \lambda_2 \Upsilon(\tilde{\mathbf{r}}_\tau) \Big\rangle \tag{2}$$

$$s.t. \quad \begin{aligned} \tilde{\mathbf{r}}_\tau &= D_\tau(\mathbf{z}) \\ \tilde{\mathbf{r}}_\tau &\in \mathcal{S}(\mathbf{a}) \end{aligned}$$

where $\Psi : [0,1]^{L \times 64} \to \mathbb{R}$ and $\Upsilon : \Psi : [0,1]^{L \times 64} \to \mathbb{R}$ are the expression and MFE predictors with $\lambda_1$ and $\lambda_2$ regulating their relative contribution respectively. $D_\tau$ is the latent-to-sequence decoder and $\mathcal{S}(\mathbf{a}) = \mathcal{C}(a_1) \times \cdots \times \mathcal{C}(a_L)$ is the space of all synonymous sequences for the given protein $\mathbf{a}$.

In the remainder of the section, we detail all of the components of the LSCO and the constrained decoding procedure to satisfy the protein consistency $\tilde{\mathbf{r}}_\tau \in \mathcal{S}(\mathbf{a})$ constraint.

**Mapping discrete sequence space to continuous codon-latents** We represent the mRNA sequence in a continuous codon-latent space $\mathcal{Z}_L$, using a decoder network $D_\tau : \mathcal{Z}_L \to \mathbb{R}^{L \times 64}$ that maps a latent vector $\mathbf{z} \in \mathcal{Z}_L$ to a probability distribution over codon identities. This mapping is achieved via a final softmax layer with temperature $\tau$. We keep the sequence dimension explicitly disentangled, as the number of codons is fixed for a given protein. The temperature $\tau$ controls the level of continuous relaxation; as $\tau \to 0$, it recovers the discrete optimization setting.

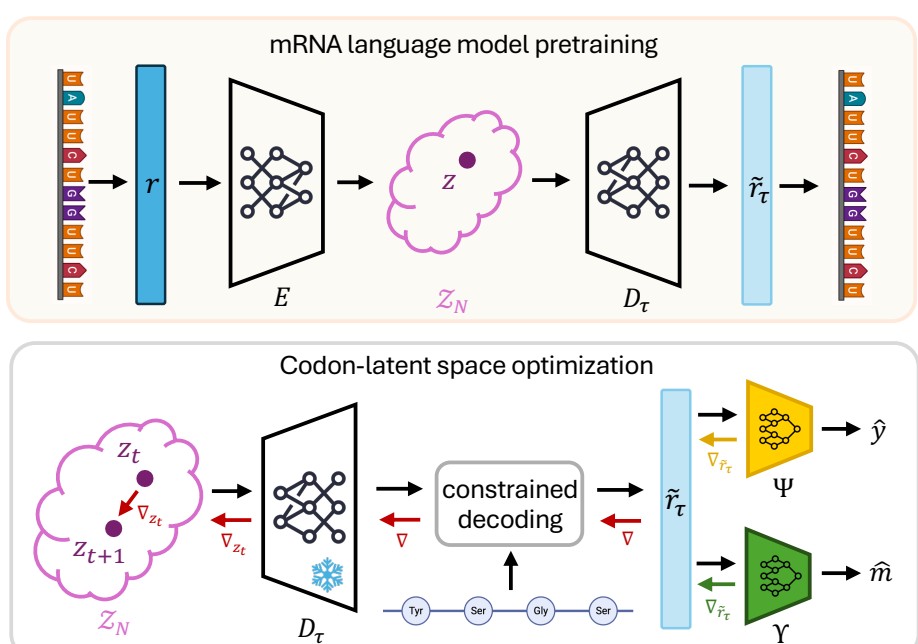

Figure 2: **LSCO framework.** (a) mRNA sequences are mapped to codon-latent space using a pretrained language model. (b) Latent representations $\mathbf{z}$ are optimized with respect to (i) a data-driven expression predictor $\Psi$ and (ii) an MFE predictor $\Upsilon$. (c) Constrained decoding ensures valid codon choices, preserving the target protein sequence. The result is an optimized mRNA that balances expression and stability.

We train $D_\tau$ as part of a BERT-based mRNA language model, treating the first $k$ layers as an encoder $E : \mathbb{R}^{L \times 64} \to \mathcal{Z}_L$ and the remaining layers as the decoder $D_\tau : \mathcal{Z}_L \to \mathbb{R}^{L \times 64}$. We select a bidirectional BERT architecture because mRNA structures enable non-causal interactions that can strongly affect the molecule's physical properties. We train the model using the standard Masked Language Modeling (MLM) objective on the OAS dataset (Yazdani-Jahromi et al., 2025).

**Data-driven expression objective** We train an mRNA-to-protein expression prediction network $\Psi : [0, 1]^{L \times 64} \to \mathbb{R}$ to guide the codon optimizer. The network takes $\tilde{\mathbf{r}}_\tau$ as input, which provides a non-Dirac delta distribution for each codon position. Thus, $\tilde{\mathbf{r}}_\tau$ acts as a soft, continuous version of the discrete mRNA sequence $\mathbf{r}$. We deliberately operate on this soft sequence representation rather than directly on the codon-latent space $\mathcal{Z}_L$. This choice stems from early experiments showing that gradient-based optimization solely in the latent space can cause large shifts in predicted expression and latent values without corresponding changes at the sequence level. Additionally, working at the soft sequence level enables a simple constrained decoding process to enforce the protein consistency constraint $\tilde{\mathbf{r}}_\tau \in \mathcal{S}(\mathbf{a})$.

We implement $\Psi$ as a sequence transformer with pooling over the sequence dimension followed by a linear layer on top. In practice, we found that reusing the pretrained language model encoder $E$ as the backbone for $\Psi$ improves the expression prediction. To enable this, we replace the first `nn.Embedding` layer of $E$ with a dense `nn.Linear` layer to handle soft non-one-hot sequences. Once trained, $\Psi$ is differentiable with respect to the soft-sequence inputs $\tilde{\mathbf{r}}_\tau$, allowing gradient-based optimization.

**Minimum Free Energy objective** To prevent near-adversarial sequences and ensure mRNA stability, we train a Minimum Free Energy prediction network $\Upsilon : [0, 1]^{L \times 64} \to \mathbb{R}$ to regularize the expression prediction objective. Lower MFE values indicate more stable secondary structures, which enhance mRNA longevity and translational efficiency in the host cell (Higgs, 2000). Like

$\Psi$, $\Upsilon$ operates on the soft sequences $\tilde{\mathbf{r}}_\tau$ for continuous differentiability and constrained decoding to maintain $\tilde{\mathbf{r}}_\tau \in \mathcal{S}(\mathbf{a})$.

We implement $\Upsilon$ similarly to the expression prediction model, as a sequence transformer with pooling and a final linear layer, reusing the pretrained encoder $E$ (with its embedding layer swapped for a dense linear one to process non-one-hot inputs) as the initial feature extractor. Our MFE prediction model is trained on MFE labels computed via established ViennaRNA (Lorenz et al., 2011), enabling it to approximate folding energies rapidly without explicit biophysical simulations. The $\Upsilon$ network is differentiable with respect to $\tilde{\mathbf{r}}_\tau$, enabling gradient-based regularization during optimization.

**Constrained decoding**  Finally, we enforce that optimization in the codon-latent space always decodes to valid mRNA sequences that translate to the target protein $\mathbf{a}$. Let $\mathbf{l} = D'(\mathbf{z}) \in \mathbb{R}^{L \times 64}$ denote the pre-softmax logits output by the decoder network (excluding the final softmax layer). For each amino acid position $a_j$, we define a validity mask

$$M(a_j)[i] = \begin{cases} 0, & \text{if codon } i \in \mathcal{C}(a_j) \text{ is valid for } a_j, \\ -\infty, & \text{otherwise,} \end{cases} \tag{3}$$

where $\mathcal{C}(a_j)$ is the set of synonymous codons encoding $a_j$. During decoding, we compute masked logits as

$$l_{j,i} \leftarrow l_{j,i} + M(a_j)[i] \tag{4}$$

for each position $j$ and codon $i$, effectively setting the logits of non-synonymous codons to $-\infty$. We then apply the softmax with temperature $\tau$ to obtain the probability distribution $\tilde{\mathbf{r}}_\tau = \text{softmax}_\tau(\mathbf{l})$. This constrained decoding ensures that every decoded sequence $\tilde{\mathbf{r}}_\tau \in \mathcal{S}(\mathbf{a})$ encodes exactly the target amino acid sequence, while allowing the optimizer to explore synonymous codon choices freely within each position.

In practice, we also observed the importance of carefully selecting the decoding temperature $\tau$. A temperature that is too high enables less constrained exploration of the synonymous sequence space but can introduce significant rounding errors when mapping $\tilde{\mathbf{r}}_\tau$ to the nearest discrete mRNA sequence $\mathbf{r}$ via argmax. Conversely, a temperature that is too low hinders exploration of the soft-sequence space by creating high-energy barriers between sequences. Empirically, we found that linearly annealing $\tau$ from a high to a low value during optimization strikes the best balance. This strategy allows the optimizer to first explore the domain unconstrainedly to identify promising candidates and then commit to one at the end of the process.

We provide a pseudocode for the LSCO in Algorithm 1.

## 4.1 IMPLEMENTATION DETAILS

**mRNA language model**  We employ a standard 12-layer ModernBERT (Warner et al., 2024), equipped with Rotary Position Embeddings (Su et al., 2024). Interestingly, experiments with smaller models (e.g., 6 layers) revealed that the LSCO optimizer struggles to jointly optimize for both expression and MFE, likely due to insufficient representational capacity of the latent space. The model uses a vocabulary of 68 tokens (64 codons plus special tokens [CLS], [SEP], [MASK], and [PAD]), with a hidden dimension of 768, 12 attention heads per layer, and an intermediate feed-forward size of 3072. We train it using the standard Masked Language Modeling (MLM) objective with a masking ratio of $p = 0.15$ for 40 epochs on the curated OAS-mRNA dataset (Yazdani-Jahromi et al., 2025), which comprises over 15 million mRNA sequences. Optimization is performed with AdamW (Loshchilov & Hutter, 2017), a batch size of 256, an initial learning rate of $1 \times 10^{-4}$ with linear warmup over the first 10% of steps followed by cosine decay. We then slice the pretrained model at layer $k = 6$, using the first half as the encoder $E$ and the latter as the decoder $D_\tau$.

**Expression predictor**  We construct the expression predictor $\Psi$ atop the pretrained encoder $E$, incorporating mean pooling across the sequence dimension and a linear regression head. The input embedding layer is swapped for a dense linear projection to support soft codon distributions. Training proceeds for 70 epochs with MSE loss on the Ab1 and Ab2 expression (absorbance measured at 280 nm) data (Prakash et al., 2024; Yazdani-Jahromi et al., 2025). We use AdamW optimization with a batch size of 64, initial learning rate of $5 \times 10^{-5}$, and the same warmup and decay schedule as the language model. The Spearman's rank correlation between the predicted and ground truth expression on the test split reaches 0.83.

---

**Algorithm 1:** Latent-Space Codon Optimizer (LSCO)

---

**Require:** Protein sequence $\mathbf{a}$, pretrained encoder $E$, decoder $D'$, predictors $\Psi$ and $\Upsilon$, weights $\lambda_1, \lambda_2$, initial $\tau_{\text{high}}$, final $\tau_{\text{low}}$, optimization steps $T$, learning rate $\eta$
**Ensure:** Optimized mRNA sequence $\mathbf{c}^\star$
 1: Initialize $\mathbf{z} \in \mathcal{Z}_L$    $\triangleright$ randomly from $\mathcal{N}(0,1)$
 2: **for** $t = 1$ to $T$ **do**
 3:    $\tau \leftarrow \tau_{\text{high}} + (t/T) \cdot (\tau_{\text{low}} - \tau_{\text{high}})$    $\triangleright$ temperature annealing
 4:    $\tilde{\mathbf{r}}_\tau \leftarrow \text{softmax}_\tau(D'(\mathbf{z}) + M(\mathbf{a}))$    $\triangleright$ constrained soft sequence
 5:    $\mathcal{L} \leftarrow -\lambda_1 \Psi(\tilde{\mathbf{r}}_\tau) + \lambda_2 \Upsilon(\tilde{\mathbf{r}}_\tau)$
 6:    Update $\mathbf{z}$ via gradient descent: $\mathbf{z} \leftarrow \mathbf{z} - \eta \nabla_{\mathbf{z}} \mathcal{L}$
 7: **end for**
 8: $\mathbf{r} \leftarrow \arg\max(D'(\mathbf{z}) + M(\mathbf{a}))$    $\triangleright$ discrete one-hot mRNA
 9: $\mathbf{c}^\star \leftarrow$ map $\mathbf{r}$ to codon sequence
10: **return** $\mathbf{c}^\star$

---

**MFE predictor**   Analogously, the MFE predictor $\Upsilon$ mirrors $\Psi$'s architecture. It is trained for 50 epochs using MSE loss on 50,000 sub-sampled OAS mRNA sequences, with ground-truth MFE labels computed using the ViennaRNA package. Optimization follows identical settings to $\Psi$. The Spearman's rank correlation between the predicted and ground truth MFE on the test split reaches 0.91.

**LSCO hyperparameters**   We determined the weights $\lambda_1$ and $\lambda_2$ via grid search on each dataset to maximize predicted A280 while ensuring the optimized sequence's MFE is generally lower than the wild-type MFE, with flexibility to relax this constraint when predicted A280 $\leq 1$. In practice, LSCO proved robust to a wide range of these weights. Grid search identified $\lambda_1 = 10$ and $\lambda_2 = 1.5$ as effective for both Ab1 and Ab2 datasets. For decoding, we set the initial temperature $\tau_{\text{high}} = 3$ and the final temperature $\tau_{\text{low}} = 0.01$. These temperature parameters showed higher sensitivity, particularly $\tau_{\text{low}}$; excessively high values caused significant rounding errors, reducing expression, while overly low values hindered optimization progress. We ran LSCO for $T = 6000$ iterations, observing that additional iterations improved convergence marginally. The learning rate was set to $\eta = 10^{-4}$, and, leveraging the differentiability of the LSCO objective, we employed the Adam optimizer for robust gradient-based updates.

## 5 EXPERIMENTS

**Baselines**   We benchmark our Latent Space Codon Optimization against other publicly available codon optimization techniques. These include frequency-based host-preferred codon selection (Sharp & Li, 1987), a maximum Ab-naturalness back-translation model a learning-based ICOR optimizer Jain et al. (2023), and the recent state-of-the-art CodonTransformer (Fallahpour et al., 2025) optimizer. To better understand the degree of contribution of each method, we also employ random synonymous codon substitution, treating it as a lower bound in terms of optimized sequence quality.

**Evaluation**   To evaluate codon optimization, we use 21 holdout sequences from Ab1 and 21 from Ab2, excluded from model training and validation. The holdout sequences were selected to minimize the overlap with the training and validation data in terms of maximum sequence similarity. We assess performance using multiple complementary RNA sequence metrics. We evaluate predicted protein expression with our trained $\Psi$ predictor, estimating translation efficiency. We quantify thermodynamic stability by the minimum free energy of the secondary structure, computed with ViennaRNA. We measure GC content to assess host compatibility, as extreme levels can reduce translation efficiency (Kudla et al., 2009). Finally, we quantify sequence naturalness as the log-likelihood under the OAS back-translation model.

### 5.1 LATENT-SPACE CODON OPTIMIZATION MAXIMIZES ANTIBODY EXPRESSION

We evaluate LSCO against baseline codon optimization methods on holdout sequences from the Ab1 and Ab2 datasets. As shown in Figures 3a and 4a, LSCO consistently produces sequences with

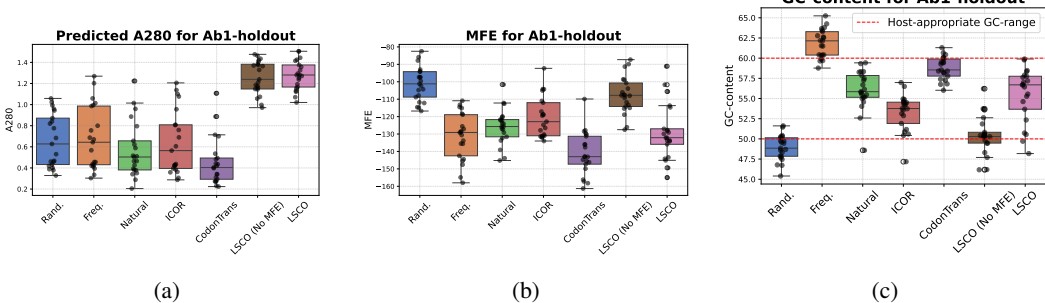

Figure 3: Codon optimizers comparison on Ab1-holdout data: (a) Predicted A280 by codon optimizer, where LSCO clearly outperforms the rest of the methods, yielding the most expressive sequences; (b) MFE by codon optimizer; all methods besides random substitution baseline produce stable sequences; (c) GC-content by codon optimizer, where random- and frequency substitution baselines largely fall out of appropriate GC-range for the host organism.

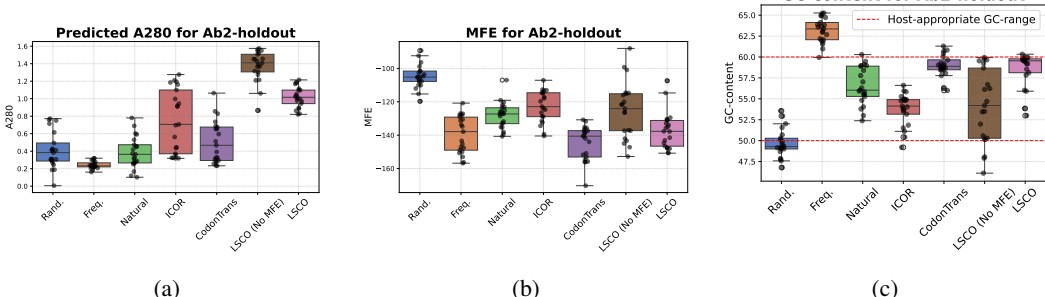

Figure 4: Codon optimizers comparison on Ab2-holdout data: (a) A280 predictions for the codon optimizers; LSCO produces sequences with the higest expression level (b) MFE evaluation indicates that the majority of methods, excluding the random substitution, generate mostly stable structures; (c) GC-content levels stay in the admissible range for most of the methods with the random and frequency substitution baselines straying beyond the admissible GC boundaries.

the highest predicted A280 expression values. It substantially outperforms other codon optimization methods, even the data-driven ICOR and the most recent CodonTransformer that rely on naturalness based proxies. This improvement stems from LSCO's ability to directly optimize for protein expression in a continuous codon-latent space as opposed to optimizing expression surrogates, which do not necessarily lead to the expression-optimal sequence. Notably, LSCO achieves high predicted expression without sacrificing structural stability, as seen in Figures 3b and 4b where LSCO-optimized sequences maintain similar stability levels to other methods.

To additionally explore the role of structural stability, we analyzed sequence GC content (Figures 3c and 4c) to check if low MFE arises from inflated GC content. High GC content drives MFE lower because GC pairs form stronger bonds (three hydrogen bonds versus two for AU pairs), leading to more stable RNA secondary structures with lower free energy; however, this can hinder protein expression if GC content exceeds the host-appropriate range, as extreme GC-content levels can impede ribosome accessibility or trigger immune responses Gustafsson et al. (2004). The GC content analysis reveals that frequency-based and CodonTransformer baselines indeed achieve low MFE at the cost of inflated GC content, pushing it outside the admissible range. This explains their poor expression. LSCO-optimized sequences, however, maintain admissible GC content, so their enhanced stability does not stem from GC inflation and better supports high protein expression.

Overall, our results show that LSCO jointly optimizes for expression, structural stability, and host compatibility, a balance that previous codon optimization methods cannot achieve.

**MFE ablation study**    To assess the role of the MFE regularization in expression-driven codon optimization, we conducted an ablation study by excluding the MFE term. Results for both the Ab1 and Ab2 datasets are summarized in Figures 3 and 4 (LSCO No MFE). For predicted expression (A280), the ablated LSCO No MFE variant achieved levels comparable to the full LSCO model. However, removing MFE substantially altered other sequence properties: GC-content distributions deviated from host-appropriate ranges, and predicted MFE values became significantly worse, reflecting reduced structural stability. These findings demonstrate that while the MFE objective does not directly enhance expression, it serves as an essential regularizer, promoting biophysical plausibility by maintaining balanced GC content and stable folding.

**Naturalness as a proxy of expression**    In Appendix A.2, we conduct additional analyses for naturalness-based codon optimization, demonstrating that naturalness only weakly correlates with actual protein expression, thus necessitating explicit expression maximization as performed in LSCO.

## 6    DISCUSSION

In this work, we introduced the Latent-Space Codon Optimizer, a novel approach that addresses the core challenges of codon optimization by shifting from discrete, heuristic-driven searches to continuous, data-driven gradient-based optimization. Unlike traditional methods that rely on proxy metrics such as CAI or GC content, which often fail to fully capture the complexities of protein expression, LSCO leverages a pretrained mRNA language model's latent space to enable efficient exploration of synonymous sequences. By directly learning an expression predictor from data and regularizing with a Minimum Free Energy, LSCO produces mRNA designs that not only maximize predicted protein yield but also maintain structural stability and host-appropriate GC levels. Our experiments on protein antibody sequences demonstrate that LSCO outperforms established baselines like frequency-based substitution, ICOR, and CodonTransformer, achieving superior expression without compromising critical sequence properties.

**Limitations and Future work**    Despite these advances, LSCO has several limitations. First, its performance depends heavily on the quality and diversity of the training data for both the language model and predictors. Second, the computational overhead of training the predictors and performing latent-space optimization could be prohibitive for very long sequences or high-throughput applications. A broader challenge in the field is the lack of publicly available datasets with comprehensive annotations for RNA expression efficiency, where measurements are extremely scarce and often proprietary. This scarcity makes it difficult to comprehensively benchmark various codon optimization methods beyond surrogate metrics, and, like many studies in this area, our experiments are limited in scale and rely on predicted rather than empirical expression data. Additionally, the lack of wet-lab validation means that predicted improvements in expression and stability remain unconfirmed in vivo.

## REPRODUCIBILITY STATEMENT

To facilitate reproducibility, we provide the important implementation details, such as model hyperparameters and training details, in Section 4.1. The baseline methods are run in the respective default configuration with the appropriate host-organism.

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

# A  APPENDIX

## A.1  LLMs USAGE

Large Language Models (LLMs) were used to check the grammar and improve the plotting code.

## A.2  NATURALNESS WEAKLY CORRELATE WITH ACTUAL PROTEIN EXPRESSION

Due to the scarcity of publicly available mRNA-protein expression data, sequence-naturalness-based metrics have become the standard proxy for expression. To examine the contribution of natural-ness as a valid expression metric, we compute the log-likelihood of mRNA sequences under the Ab-specific back-translation model and correlate these scores with experimentally measured A280 values on the Ab1 and Ab2 datasets. On Ab1, the correlation between naturalness and expression is weak ($r = -0.28$), while on Ab2 it is negligible ($r = 0.04$), as shown in Figures 5a and 5b. These results indicate that sequences optimized purely for naturalness do not reliably yield high protein expression. High-expression sequences can appear unnatural, and low-expression sequences can be highly natural, underscoring the limitations of naturalness as a proxy for protein expression.

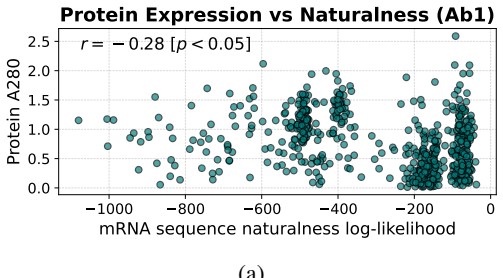 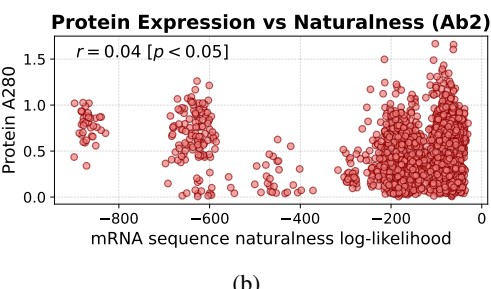

(a)                                          (b)

Figure 5: Experimentally measured protein A280 vs. mRNA naturalness log-likelihood: (a) on Ab1 dataset, with weak correlation ($r = -0.28$); (b) on Ab2 dataset, with virtually no correlation ($r = 0.04$).

