# OpenReview forum: "Latent Space Codon Optimization Maximizes Protein Expression"
_ICLR.cc/2026/Conference — ICLR 2026 Conference Withdrawn Submission_

### Official Review · Reviewer_u2T9 · 2025-10-25

**Soundness:** 2
**Presentation:** 3
**Contribution:** 2
**Rating:** 2
**Confidence:** 5

**Summary:**

The paper introduces Latent-Space Codon Optimizer (LSCO) to tackle codon optimization’s two core challenges, an enormous discrete search space and the inadequacy of proxy objectives like CAI/GC to capture true expression, by reframing the problem as continuous, gradient-based optimization guided by learned objectives. LSCO operates in a pretrained mRNA language model’s latent space and uses a differentiable expression predictor regularized by a learned Minimum Free Energy (MFE) model to promote structural stability and avoid proxy-gaming. Decoding is constrained with positionwise masks so the final sequences strictly preserve the target amino-acid sequence, with temperature annealing balancing exploration and commitment. Evaluated on antibody datasets (Ab1/Ab2), the authors shw that LSCO achieves the highest predicted expression while maintaining stability and host-appropriate GC ranges, outperforming frequency-based methods, ICOR, and CodonTransformer; ablations show the MFE term acts as a crucial biophysical regularizer. They also do note limitations around data scarcity, compute demands, and the absence of wet-lab validation, which leaves improvements empirical-prediction–based rather than experimentally confirmed.

Overall, the paper is not well-suited for an AI conference. I like the paper overall as an application work, but it doesn't suit ICLR. The work would require extensive wet lab validation for it to be a publishable paper at any venue. I don't think the authors will have time for that during the rebutal period.

**Strengths:**

1. I like that the paper converts codon optimization from intractable discrete search with weak proxies into a continuous, gradient-based optimization in latent space, directly addressing the two core challenges.
2. Experiments indicate that LSCO improves predicted expression while maintaining stability (via MFE), which is practical on the reported datasets. However, this is pretty meaningless with experimental data.

**Weaknesses:**

1. Optimizing discrete sequences via latent/soft representations is a well-established paradigm. I am inclined to see this contribution as an application rather than a new optimization framework. Thus, it may be more suitable for a journal (say Nature Machine Intelligence or Nature Computational Science)
2.  Although the paper frames the task as continuous multi-objective optimization, it does not compare against strong discrete multi-objective methods (e.g., (i) https://arxiv.org/abs/2412.17780 (ii) https://arxiv.org/abs/2505.07086 (iii) https://arxiv.org/abs/2510.00352), which makes it unclear to mewhether mapping to a continuous domain is necessary or superior.
3. The authors focus on antibody datasets; but broader organisms, cell types, and sequence classes were not tested. It's not the authors fault, per say, but the acknowledged scarcity of public expression datasets definitely limits both generality and head-to-head comparisons.
4. The evaluation uses only 21 holdouts per dataset (Ab1/Ab2), which makes me a bit concerned about reliability, variance, and potential bias of the reported gains.

**Questions:**

1. What is the symbol `E` referenced in line 166? Is this a notation typo that should be $\Phi$?
2. Beyond linear scalarization, the authors shoukld try Tchebycheff or ϵ-constraint methods. Since this is a multi-objective problem, I'd like to see how well does your method cover the Pareto front (e.g., hypervolume/coverage metrics, front shape)?
3. Why did the authors rely on grid search for $\lambda_1$ and $\lambda_2$ instead of sampling weights on the probability simplex (for example. a $\Delta$-lattice or Dirichlet over $\lambda$) to encourage broader Pareto coverage? It's been established that grid search can become expensive and brittle when additional property constraints are introduced.

**Details Of Ethics Concerns:**

None.

---

> ### Author Response · Authors · 2025-11-30
>
> We thank the reviewer for the detailed feedback, which will be very useful for improving the next version of our work.
>
> - We agree that the evaluation is limited, and we fully agree that evaluating on more datasets and more diverse proteins would strengthen the conclusions. At the moment, there are very few public datasets that provide both mRNA sequences and matched expression measurements at a suitable scale and variety. As new datasets appear, we plan to incorporate them to broaden the evaluation.
>
> - On comparisons to PepTune [1], MOG-DFM [2], and AReUReDi [3]: We appreciate the suggestions for additional related work. PepTune and MOG-DFM guide and refine generative models (e.g., flow matching). These work best when the base unconditional model already produces high-likelihood samples that need steering toward high-likelihood regions with desired properties. In codon optimization, however, high-likelihood sequences are not always high-expression ones; in fact, the opposite is often true (see Fig. 5 and the Natural baseline in Figs. 3 and 4). This makes sense, as large mRNA datasets suitable for pretraining a generative model typically contain natural sequences, while codon-optimized sequences can be highly unnatural. Starting from a generative model and then steering it is thus challenging without large databases of both natural and unnatural (optimized) mRNA sequences.
>
> For AReUReDi [3], our paper was submitted on September 24, 2025, while AReUReDi first appeared on arXiv on September 30, 2025, which is five days after our submission. Not only was it impossible for us to compare with AReUReDi at the time of submission, but also the official ICLR guideline clearly defines concurrent/contemporaneous work as the work published within the last four months before submission of the paper.
>
> Additional questions:
> - The symbol E in line 166 is a typo, and should be $\Phi$ from Eq.(1).
> - We appreciate the reviewer's suggestion on Tchebycheff or ϵ-constraint, and we will additionally evaluate these scalarization methods in future versions of the work.
> - We thank the reviewer for suggesting alternative strategies for sampling weights for the optimization objective. We will ablate various strategies of setting lambdas in future versions of the work.
>
> [1] Tang, Sophia, Yinuo Zhang, and Pranam Chatterjee. "Peptune: De novo generation of therapeutic peptides with multi-objective-guided discrete diffusion." ArXiv (2025): arXiv-2412.
>
> [2] Chen, Tong, et al. "Multi-objective-guided discrete flow matching for controllable biological sequence design." arXiv preprint arXiv:2505.07086 (2025).
>
> [3] Chen, Tong, Yinuo Zhang, and Pranam Chatterjee. "AReUReDi: Annealed Rectified Updates for Refining Discrete Flows with Multi-Objective Guidance." arXiv preprint arXiv:2510.00352 (2025).

---

### Official Review · Reviewer_ETR1 · 2025-10-31

**Soundness:** 3
**Presentation:** 2
**Contribution:** 2
**Rating:** 2
**Confidence:** 4

**Summary:**

The paper proposes LSCO, which reformulates codon optimization, a discrete combinatorial search over synonymous mRNA sequences, as a continuous gradient-based optimization problem in the latent space of a pretrained mRNA language model.

**Strengths:**

-The challenge in codon optimization is the discrete, combinatorial search space. LSCO's primary strength is converting this into a continuous latent space using a language model.

-The paper includes an ablation study (LSCO No MFE) that removes the MFE regularization. The results show that without MFE, the model produces unstable sequences with bad GC content.

**Weaknesses:**

-Only two antibody datasets (Ab1, Ab2) are used, with <50 total holdout sequences. No cross-species or variable-length test cases are shown.insufficient diversity for claims of “general therapeutic applicability.”

-the expression predictor reuses the encoder E from the pretrained LM trained on the same distribution, making representation leakage likely between training and evaluation.

-The gradient step in latent space assumes local smoothness — that small Δz corresponds to small Δsequence and Δexpression. But codon choices are often non-locally coupled (due to RNA secondary structure). There’s no proof that the latent manifold preserves this continuity.

-it seems that no source code provided, private validation dataset...

-would like to improve my score if above concerns can be well addressed

**Questions:**

-How robust are the optimized sequences when re-evaluated by an independent expression predictor not used in optimization?

-Why not comparing with LinearDesign? (previous SOTA

-why  using real experimental data to train the expression predictor ($\Psi$), but then use that same predictor, and not real experimental data, to evaluate the performance of the final method (LSCO)? This doesn't seem to make sense.

---

> ### Author Response · Authors · 2025-11-30
>
> We thank the reviewer for the detailed feedback on weaknesses, which will be very useful for improving the next version of our work.
>
> - We agree with the reviewer that the experimental evaluation is limited, and we fully agree that evaluating on more datasets and more diverse proteins would strengthen the conclusions. At the moment, there are very few public datasets that provide both mRNA sequences and matched expression measurements at a suitable scale and variety. As new datasets appear, we plan to incorporate them to broaden the evaluation.
>
> - Regarding the leakage between training and evaluation, this is a valid concern. However, note that the optimization is done in a continuous latent space, while mRNA sequences are discrete, and hence the evaluation is performed in the discrete sequence space as well. Any optimized point must be decoded and rounded back to a valid mRNA sequence. This makes constructing purely adversarial sequences more difficult in practice, since many small changes in the latent space do not translate into arbitrary changes in the final discrete sequence. This considerably reduces the risk of just "hacking" the expression predictor.
>
> - Regarding the local smoothness assumption for gradient steps in latent space: we agree that this is a key modeling assumption and that codon choices can interact non-locally through RNA secondary structure. This limitation is not unique to our setting; it is shared by many continuous relaxations of combinatorial problems. Without some form of smoothness assumption, one is forced back into full combinatorial optimization, e.g., dynamic programming methods, which bring their own scalability issues. While we agree that the local smoothness in modeled expression and MFE may not fully match the discrete landscape, one of the core contributions and insights of the LSCO is that it demonstrates that the optimization in such continuous space still yields valid, highly-expressive mRNA sequences.

---

### Official Review · Reviewer_NHrp · 2025-11-01

**Soundness:** 2
**Presentation:** 3
**Contribution:** 2
**Rating:** 4
**Confidence:** 3

**Summary:**

This paper introduces Latent-Space Codon Optimization that optimizes mRNA codon sequences in a continuous latent space obtained from a pretrained mRNA BERT models. The optimization is guided by the gradients of estimated mRNA protein expression, further regularized with a Minimum Free Energy for structure stability. They also leveraged constrained decoding to ensure the decoded mRNA sequence codes the original protein. In empirical experiments, they showed LSCO outperforms other baselines on optimizing protein expression while also maintaining solid metrics on MFE and GC, striking a better balance between protein expression optimization, structural stability, and GC levels than the other existing methods.

**Strengths:**

1. The paper is well organized and flows naturally. I appreciate the comprehensive introduction to the mRNA codon optimization task.
2. On empirical validation, they demonstrated that the proposed method LSCO is able to achieve much better protein expression values and is also structurally stable and maintains good GC levels compared to other baselines.
3. The temperature annealing for the constrained decoding during optimization is clever.

**Weaknesses:**

1. I am concerned that the same expression predictor is used both to optimize LSCO and to evaluate performance. This poses a risk of circularity or overfitting to the predictor model rather than truly improving expression.

2. Evaluation is limited to 21 holdout sequences each from Ab1 and Ab2. This is a very small and single source test set; it is unclear if this generalizes to other proteins or settings.

3. The ablation study could be more interesting by studying more components of the proposed network in addition to studying with and without MFE. For example, in the paper, it’s stated that the authors adapted one design choice over the other such as annealing the temperature vs. setting a fixed one, optimizing in latent space vs. soft sequence space.

4. Lack of qualitative analysis. I appreciate the analysis of the relation between MFE and GC levels for the quantitative analysis but there is no qualitative analysis. For example, it would be interesting to see the latent space used in optimization.

5. Novelty-wise, while latent-space optimization for biological sequences is not entirely new, its application to codon optimization is interesting. However, the paper could more clearly distinguish itself from prior latent-space molecular design works.

**Questions:**

I am curious what’s the success rate is for constraining/preserving the decoded sequences, since it’s not discussed or reported in the paper.

While I find the proposed latent-space approach to codon optimization well-motivated and well-presented, I am concerned that the evaluation does not convincingly demonstrate biological improvement. The same expression predictor is used both for optimization and for evaluation, raising concerns of circularity. Moreover, the experiments dataset is small and coming from a single source. As a result, I am not confident in the strength or generality of the reported gains. I therefore lean toward weak reject. I believe the paper has potential and could be strengthened with independent evaluation and more experimental validation.

---

> ### Author Response · Authors · 2025-11-30
>
> We appreciate the reviewer highlighting that our method is well-motivated and well-presented, and we thank the reviewer for the detailed feedback on weaknesses, which will be very useful for improving the next version of our work.
>
> - Regarding the use of the same predictor at train and evaluation time (circularity), we point out that the optimization is done in a continuous latent space, while mRNA sequences are discrete. Any optimized point must be decoded and rounded back to a valid mRNA sequence. This makes constructing purely adversarial sequences more difficult in practice, since many small changes in the latent space do not translate into arbitrary changes in the final discrete sequence. This considerably reduces the risk of just "hacking" the expression predictor.
>
> - Novelty-wise, we want to highlight that the proposed codon optimization method is, to our knowledge, the first in-silico method to directly target protein expression maximization, rather than only optimizing proxies such as CAI, MFE, or sequence “naturalness” for mRNA design. In future versions of the paper, we will add a more detailed related work paragraph on latent-space molecular design and clearly discuss how our method differs.
>
> - We agree that the evaluation is limited, and we fully agree that evaluating on more datasets and more diverse proteins would strengthen the conclusions. At the moment, there are very few public datasets that provide both mRNA sequences and matched expression measurements at a suitable scale and variety. As new datasets appear, we plan to incorporate them to broaden the evaluation.
>
> - On ablations, we agree that studying more components of the system would be valuable. In a future version, we plan to extend the ablation study beyond the MFE regularizer to include: (i) annealed versus fixed temperature schedules in the decoder, and (ii) optimization in latent space versus a soft sequence-space relaxation. This will make clearer which design choices are most important for performance.
>
> Additional questions:
> - The constrained decoding procedure is designed to always yield a valid mRNA sequence that strictly corresponds to the target protein by the end of optimization, which is achieved by annealing the constrained decoding temperature to a low value (lines 299–300); in our experiments, we did not observe any decoding failures or violations of the protein constraint.

---

### Official Review · Reviewer_WgiX · 2025-11-01

**Soundness:** 1
**Presentation:** 2
**Contribution:** 2
**Rating:** 2
**Confidence:** 3

**Summary:**

This paper introduces LSCO, which frames codon optimization as a continuous optimization problem in the latent space of a pretrained mRNA language model. It uses a trained expression predictor and a minimum free energy (MFE) predictor to jointly improve mRNA expression and stability.

**Strengths:**

- Clearly defines the codon optimization problem, an underexplored but important problem.
- Presents a efficient framework for codon optimization that avoids exploring combinatorial space.

**Weaknesses:**

The evaluation setting is fundamentally flawed. The model is directly trained to optimize a predictor that is exactly same checkpoint used in evaluation. There are several problems associated with it.

1. All comparisons (baselines) are unfair since none of them are using this trained predictor that LSCO used. It is impossible to conclude latent-space optimization is a useful method without having a fair baseline. Example of fair baseline would be mRNA language model alignment or fine-tuning using the predictor that LSCO used.
2. Also, from the related works (Line 115) RNop seems reasonable and strong baseline, why is this excluded from comparison?
3. The validation metrics that authors provide to claim that the model is not hacking the predictor is MFE and GC content, which I found not convincing enough. Authors mention multiple metrics in line 119 (CAI, GC content, mRNA folding energy, and codon pair bias), why only report GC content? MFE is better since authors used trained predictor at train time and ViennaRNA MFE at evaluation time, but can you discuss how ViennaRNA computes MFE? That will help readers to understand how easy/hard to hack the metric.

Benchmark is very weak. While the problem of codon optimization is general, this method is evaluated in antibody dataset only, and the evaluation set has only 21 sequences.

**Questions:**

See Weaknesses.

---

> ### Author Response · Authors · 2025-11-30
>
> We thank the reviewer for the detailed feedback, which will be very useful for improving the next version of our work. Next, we address the main concerns about the LSCO evaluation.
>
> - We acknowledge limitations of the current evaluation protocol, but we do not agree that it is fundamentally flawed; rather, we view it as the first step in a more comprehensive evaluation of the LSCO. Our main goal in this work is to directly optimize predicted expression, which, to our knowledge, previous codon optimization methods do not do as they rely on indirect proxies such as MFE or sequence "naturalness”. Because of this, those baselines are exposed to similar evaluation biases, as in fact, optimizing naturalness or related proxies does not reliably lead to high expression [1,2]. This is reflected in the modest expression levels achieved by the Natural, ICOR, and CodonTransformer baselines in Figure 3, despite their strong proxy scores.
>
> - Regarding the use of the same predictor at train and evaluation time, we point out that the optimization is done in a continuous latent space, while mRNA sequences are discrete. Any optimized point must be decoded and rounded back to a valid mRNA sequence. This makes constructing purely adversarial sequences more difficult in practice, since many small changes in the latent space do not translate into arbitrary changes in the final discrete sequence. This considerably reduces the risk of just "hacking" the expression or MFE predictor.
>
> - On evaluation metrics: we do not only report the predictor score, but also MFE and GC content. Other metrics (such as CAI, codon pair bias, or DTW [3]) have been argued to be unreliable as predictors of actual protein expression [1,2], which is the primary quantity of interest in codon optimization. That said, we agree these additional metrics can still be informative and help provide a more complete picture of the designed sequences, and we will include them in a future version.
>
> - We agree that the benchmark is limited, and we fully agree that evaluating on more datasets and more diverse proteins would strengthen the conclusions. At the moment, there are very few public datasets that provide both mRNA sequences and matched expression measurements at a suitable scale and variety. As new datasets appear, we plan to incorporate them to broaden the evaluation.
>
> [1] Plotkin, Joshua B., and Grzegorz Kudla. "Synonymous but not the same: the causes and consequences of codon bias." Nature Reviews Genetics 12.1 (2011): 32-42.
>
> [2] Mauger, David M., et al. "mRNA structure regulates protein expression through changes in functional half-life." Proceedings of the National Academy of Sciences 116.48 (2019): 24075-24083.
>
> [3] Fallahpour, Adibvafa, et al. "CodonTransformer: a multispecies codon optimizer using context-aware neural networks." Nature Communications 16.1 (2025): 3205.

---

### Note · Authors · 2025-12-01

I have read and agree with the venue's withdrawal policy on behalf of myself and my co-authors.